# Inter-Individual Variability in tDCS Effects: A Narrative Review on the Contribution of Stable, Variable, and Contextual Factors

**DOI:** 10.3390/brainsci12050522

**Published:** 2022-04-20

**Authors:** Alessandra Vergallito, Sarah Feroldi, Alberto Pisoni, Leonor J. Romero Lauro

**Affiliations:** 1Department of Psychology & NeuroMi, University of Milano Bicocca, 20126 Milano, Italy; alberto.pisoni@unimib.it (A.P.); leonor.romero1@unimib.it (L.J.R.L.); 2School of Medicine and Surgery, University of Milano-Bicocca, 20854 Monza, Italy; s.feroldi@campus.unimib.it

**Keywords:** tDCS, inter-individual differences, inter-subject variability, reproducibility, noninvasive brain stimulation

## Abstract

Due to its safety, portability, and cheapness, transcranial direct current stimulation (tDCS) use largely increased in research and clinical settings. Despite tDCS’s wide application, previous works pointed out inconsistent and low replicable results, sometimes leading to extreme conclusions about tDCS’s ineffectiveness in modulating behavioral performance across cognitive domains. Traditionally, this variability has been linked to significant differences in the stimulation protocols across studies, including stimulation parameters, target regions, and electrodes montage. Here, we reviewed and discussed evidence of heterogeneity emerging at the intra-study level, namely inter-individual differences that may influence the response to tDCS within each study. This source of variability has been largely neglected by literature, being results mainly analyzed at the group level. Previous research, however, highlighted that only a half—or less—of studies’ participants could be classified as responders, being affected by tDCS in the expected direction. Stable and variable inter-individual differences, such as morphological and genetic features vs. hormonal/exogenous substance consumption, partially account for this heterogeneity. Moreover, variability comes from experiments’ contextual elements, such as participants’ engagement/baseline capacity and individual task difficulty. We concluded that increasing knowledge on inter-dividual differences rather than undermining tDCS effectiveness could enhance protocols’ efficiency and reproducibility.

## 1. Introduction

In the past two decades, transcranial Direct Current Stimulation (tDCS) emerged as a novel noninvasive brain stimulation (NIBS) technique able to modulate individuals’ cortical excitability and, in turn, cognitive behavior ([1,2]; see [3] for a revision). The benefits of tDCS, such as being cheap, safe, portable, and relatively easy to use, vastly increased its employment in research and clinical settings. TDCS has been widely used to relate neural activity with sensorimotor and cognitive functions. Moreover, it has been combined with other techniques to probe its neurophysiological mechanisms. Animal in vivo and in vitro studies and human research suggested that tDCS aftereffects rely on long-term synaptic potentiation [4,5,6,7,8]. This property fostered the use of tDCS in clinical settings as adjuvant treatment in rehabilitation protocols in neurological and psychiatric conditions (for recent systematic revisions and meta-analyses, see [9,10,11]).

TDCS delivers a weak electrical current to the brain through at least two rubber electrodes (anode and cathode) placed over the scalp [3]. This current modulates the spontaneous neural firing rate in a polarity-dependent way. Specifically, anodal tDCS transiently enhances the spontaneous neural firing rate, while cathodal decreases it [12,13]. However, the effect of the two polarities on behavior is less clear. The anode-excitatory cathode-inhibitory coupling seems to accommodate only neurophysiological measurements acquired by stimulating the sensorimotor cortices [2,14]. Conversely, participants’ involvement in cognitive functions requiring more complex cerebral networks produced mixed results regarding effectiveness and modulation direction [15]. Several meta-analyses documented this inconsistency, typically concluding that tDCS has minor or null effects [16,17] or that its effectiveness is restricted to certain outcome measures [18]. However, cautions concerning the use of meta-analyses in brain stimulation studies have been previously highlighted by prominent scholars due to the small sample sizes tested and the lack of clarity concerning inclusion and exclusion criteria ([19,20]; see [21] for a general discussion of these issues in meta-analyses). Indeed, several methodological elements may contribute to the inconsistent results at the inter- and intra-study level. The inter-studies variability has been extensively reported and discussed [22,23]. It might arise from the wide range of stimulation parameters and procedures applied in the different experiments, such as tDCS intensity and duration, current density, targeted regions, electrodes’ positioning, and the use of online vs. offline protocols.

Moreover, the targeted function and employed tasks represent a relevant source of variability. Indeed, according to some authors, tDCS effectiveness may be limited to processes linked to specific behavioral tasks rather than to broader domains [24]. Even the type of measurement can introduce some variability across datasets; indeed, previous evidence suggests that specific outcome measures may be more sensitive to tDCS modulation, such as reaction times, compared with accuracy [18,25].

Whereas significant work has been carried out in terms of examining the impact of inter-studies divergencies in the heterogeneity of the tDCS effect, there is a shortage of research assessing the considerable role of intra-studies variability, namely interindividual differences in response to tDCS. Previous studies investigating the motor cortex excitability suggested that more than 50% of tested participants may not present the typical pattern of aftereffects in response to the delivered stimulation protocols [26,27]. Participants showing the expected response have been defined as *responders*, while participants that are not delivering the expected outcome are *non-responders* ([26]; see [28] for a recent review on subgrouping categorization methods). Appendix A summarizes these works, including responders vs. non-responders’ categorization, analyses, and results description [26,27,29,30,31,32,33,34,35,36,37,38,39]. The included articles reported a significant percentage of non-responders, varying across the studies. We want to stress that the low response rate might not be based only on inter-individual differences. Conversely, discrepancies can arise from experimental features: stimulation protocols (tDCS duration and intensity, TMS intensity, the interval between tDCS stimulation and MEPs recording, the presence/absence of a sham condition), statistical procedures (e.g., cluster analysis vs. pre-post change, median split, etc.), and the number of participants included.

Similar results have been found in the working memory domain [37,38]. Interestingly, in their studies, the authors involved participants in a digit span backward [37] and an n-back and Sternberg tasks [38] that followed tDCS at two different timings, namely 5 and 10 min after stimulation. Authors found that responders’ rates ranged between 15% and 59%, varying for the specific task, the interval between the end of stimulation and task administration, and the behavioral outcome measure analyzed (accuracy vs. reaction times). These findings suggested that being a responder is not a stable and fixed property but a condition depending on several contingent factors.

Strangely enough, the literature largely neglected inter-individual differences in tDCS effects. First, results are typically reported at a group level, comparing, for example, the group receiving the real stimulation vs. the group assigned to the sham/placebo condition. These analyses have the limitation of partially masking inter-individual differences by showing only average patterns. Moreover, the impact of inter-individual differences might be even more prominent if we consider the null results publication bias, namely the low percentage of studies obtaining null effects accepted and published by journals. In this sense, it is difficult to disentangle if null results are due to bad methodological experimental design (low statistical power, lack of a proper control condition), the absence of tDCS effectiveness in the target domain/cortical region, or the presence of a low proportion of responders.

Aiming at providing a complementary contribution to this topic, in the present narrative review, we investigated the heterogeneity in tDCS results emerging from inter-individual differences within the same study rather than from inter-study variability. A first division considering individual differences has been made between non-modifiable and modifiable factors [40]. Non-modifiable factors remain stable for each participant also between consecutive sessions. Differently, modifiable factors consist of characteristics that can vary between individuals and between different sessions within the same individual. On this basis, inter-individual variability in response to tDCS can be ascribed to the following categories of factors: (i) **stable factors**, such as demographic morphological and genetic characteristics; (ii) **variable state-based factors**, such as participants’ alertness, change in the hormonal or neurochemical activity, exogenous substance or drug consumption; and, among variable factors, a third category could be identified, including (iii) **experimental contextual elements**, such participants’ engagement/baseline capacity in the performed task—which is closely linked to the task difficulty—and activities carried out before/during the stimulation according to the study design. Crucially, all these elements are thought to interact in influencing tDCS effects on participants’ performance, leading to quite complex patterns [41]. In the following paragraphs, we will review and discuss previous works concerning this topic, collecting and discussing evidence on these three factors categories influencing tDCS effects at the individual level.

## 2. Stable Individual Factors of Inter-Individual Variability

Individual stable traits include anatomical and morphological features, demographic characteristics such as gender, age, and genetic profile. The most relevant and investigated elements are the morphological characteristics, such as the individual’s skull thickness, scalp-to-cortex distance, cortex folding, neurotransmitters, and genetic profile. Much evidence suggests that differences at these levels can shape the electric field distribution over the cortex, thus stimulating brain regions in different ways [42,43,44].

### 2.1. Anatomical Features: Skull Thickness, Cortex Morphology, and Gyrification

The skull morphology affects how the tDCS electric field reaches the brain cortex [43,45,46,47,48,49,50,51,52] (for studies examples, see Appendix A—section A). A first morphological factor to be considered is skull thickness—defined as the distance between the skull’s outer surface and the cerebrospinal fluid (CSF) bone interface—which strongly determines the amount of current that can pass from the skin into the brain [43,44,45].

Electric field strength has been reported to be higher under thinner skull regions. Furthermore, bone composition also plays an essential role in defining the amount of current reaching the cortical surface. Indeed, the large proportion of higher conducting spongy bone in thicker areas could lead to a more homogenous current distribution over the skull [43]. The parietal, frontal, and occipital bones are thicker in our heads and have a significant percentage of spongy bone inside. In contrast, the temporal bones are thinner and consist of compact bone [45].

Among individuals’ anatomical differences, the distance between the scalp and the cortex is another relevant factor affecting the electric field reaching the brain. Indeed, increasing the distance between the electrodes and the gray matter reduces the current density that reaches the cortex. Moreover, this space should not be considered as an empty region since it is fulfilled with CFS, which on the one hand has high conductivity [43,53] and on the other, reduces the spatial resolution of the induced electrical field [54,55,56,57].

Although individual morphological differences seem important when considering stimulation in healthy individuals, they become far more significant when brain-damaged patients are considered. For example, Datta and colleagues [58] evaluated how electric current delivered by tDCS was modulated by skull defects characterizing patients with brain injuries or craniectomy using magnetic resonance imaging (MRI) human head models. Positioning the electrode directly over a moderate-sized skull defect led to the most prominent peak cortical electric field increase. Interestingly, minor irregularities between electrodes did not significantly change cortical currents. However, by increasing the conductivity of large skull defects/plates, the current was shunted away from the underlying cortex and concentrated in the cortex under the defect perimeter. Indeed, skull defects have different conductivity, with the scar tissue of the chronic phase being the least conductive and the CSF in the acute phase being the most conductive [43,59].

Regarding the space between the skull and cortex, some considerations must also be made concerning brain atrophy, which characterizes physiological and pathological aging [60,61].

For example, Mahdavi et al. [56] showed that decreasing gray matter volume in mild cognitive impairment (MCI) and physiological aging reduced the magnitude of the current density reaching the brain compared to younger participants. Moreover, morphology alterations of the cerebral sulci could shape the vectors of current density to flow through the CSF in the depth of cortical regions. When considering Alzheimer’s patients, the same authors [62] suggested that the anatomical alterations might shift the stimulated areas and peak current density location in the head. Changes at this level might alter the expected behavioral results from tDCS application. So, they suggested that individual patterns of brain atrophy should be considered before setting the tDCS parameter. However, opposite conclusions were offered by Unal and colleagues [63]. They failed to find a correlation between the minimal distance from the gyri/sulci to the skull and the electric field intensity and concluded that there was no specific need to adjust the tDCS montage for patients with brain atrophy.

Individual differences in anatomical cortical gyrification and fiber connectivity between brain regions can also influence current distribution in the brain [23,51,61,64,65,66,67]. The high variability in gyri and sulci patterns between individuals is likely to produce high interindividual variability in tDCS response [68,69]. Of course, it is crucial to demonstrate whether morphological differences affecting the amount and distribution of the induced electrical field might impact tDCS effects at a behavioral level. Only a few works tackled this crucial aspect. For example, Filmer and colleagues [70] demonstrated that variations in the cortical architecture predicted the extent to which anodal tDCS could modify behavioral performance. In their experiment, the authors engaged 47 healthy young adults in a decision-making task during anodal or cathodal tDCS delivered over the left pre-frontal cortex. Results suggested that individuals with thicker cortex in the middle frontal sulcus and the opercular portion of the left inferior frontal gyrus and thinner cortex in the left inferior frontal triangular gyrus showed more disruption of learning with anodal stimulation. Conversely, interindividual differences could not account for the variation in cathodal tDCS efficacy, suggesting that sources of variance for the two stimulation polarities might vary.

Moreover, selective effects of tDCS across cell types and cortical layers are largely driven by their different axonal arborization and myelination profiles [53,71,72]. Indeed, the effect of the polarizing current is not always homogenous due to the orientation of neurons [73,74]. Radial current flow seems to be most effective at causing somatic polarization, whereas tangential current flow appears to be most effective at driving terminal polarization [75,76]. So, the orientation of neurons in brain layers is also a critical determinant of the polarizing effect of tDCS.

Using computational modeling, Komarov and colleagues [77] predicted that surface-anodal current preferentially activated vertically oriented neurons directly beneath the electrode. In contrast, the cathodal current would activate brain structures in the sulci. Indeed, neurons in the deeper cortical layers are often deactivated by anodal stimulation and activated by cathodal stimulation. These effects interacted with the precise placement of the electrodes relating to gyral crowns and the bipolar stimulation used. Crucially, they predicted a much higher spiking response to anodal compared to cathodal stimulation and the existence of the optimal stimulation intensity capable of inducing a maximal response in a cortical cells’ population. However, whether and how these functional modifications dynamically affect the neurophysiological response and the behavioral outcome is still to be investigated.

### 2.2. Neurochemical Factors

The neuromodulatory effects of tDCS and long-term depression (LTD) or long-term potentiation (LTP) have a close relationship with neurotransmitter levels and neuroactive drug assumption [23,78,79]. Neurotransmitters, such as GABA, glutamate, dopamine, serotonin, and acetylcholine, alter neuroplasticity in non-linear and complex ways [42] (see Appendix A—section B). Among these, GABA is strongly related to the functionality of local cortical circuits at rest and has a close relationship with tDCS outcome [23,78,80]. Generally, anodal tDCS reduces local GABA levels, while cathodal stimulation reduces glutamatergic neuronal activity. However, due to the close biochemical relationship between these two neurotransmitters, it has been found that both anodal and cathodal tDCS could decrease GABA levels [78,80].

There is an optimal amount of GABA and dopamine for brain performance [78], and this may account for some unexpected effects of tDCS at the group level [27,81]. Similarly, the facilitatory effect of anodal tDCS on the motor cortex seems to depend on the baseline activity of dopamine receptors [82,83].

These findings are even more relevant for neurological and psychiatric diseases where these neuromodulators are often out of balance [42]. Thus, studying the contribution of local neurotransmitters, mostly GABA and dopamine, may help recognize their “optimal” level to reach the full potential of tDCS.

### 2.3. Genetic Profile

Individuals’ genotype influences neuroplasticity and neurotransmitters expression, consequently impacting neurostimulation effects. This relation was first established for transcranial magnetic stimulation (TMS) and then confirmed in tDCS protocols [23,46] (see Appendix A—section C for studies examples). Indeed, the individual genetic profile seems to modulate the chance of inducing behavioral effects through tDCS and, eventually, their magnitude and duration. The brain-derived neurotrophic factor (BDNF) gene single nucleotide polymorphism is the most investigated factor influencing individuals’ response to tDCS. For instance, a seminal work by Fritsch and colleagues [84] showed that the LTP usually induced by direct current stimulation in mice motor cortex slices is absent in BDNF and TrkB mutant mice. These results suggest that BDNF is a crucial mediator of (t) DCS-induced LTP, hence the possibility to induce LTP affecting synaptic plasticity.

Healthy subjects can be grouped into individuals without the BDNF polymorphism (Val/Val) and with the BDNF single nucleotide polymorphism (Met Carrier: Val66Met or Met66Met).

In animals, the Val66Met polymorphism has been associated with synaptic plasticity abnormalities. Humans with Val66Met or Met66Met BDNF polymorphism showed only late facilitation of MEPs amplitude after anodal tDCS on the motor cortex [85].

Genotype differences can alter the tDCS effect, influencing individuals’ anatomical and neurophysiological states [23]. For example, Met BDNF polymorphism carriers showed differences in regional brain volumes and task-specific synchrony on EEG, which predicted performance on an error-processing task [86]. Moreover, the Val66Met polymorphism has been linked to a reduction in glutamatergic transmission, such that longer stimulation durations (20 min) are required to induce neuroplastic effects and achieve maximal benefits [33]. A trend towards altered plasticity in Met carriers was also observed by Antal and colleagues [87] regarding both tDCS polarity compared with the response of Val66Val individuals; however, they found no significant interaction of time on genotype.

The catechol-O-methyltransferase (COMT) gene polymorphism has also been explored. The COMT Val/Met polymorphism modulates the impact of tDCS applied to the left dorsolateral prefrontal cortex during the administration of tasks based on executive function, as the Go/No-Go paradigm. Indeed, cathodal tDCS impaired response inhibition in COMT Val/Val homozygous but not in Met-carriers [88]. In another work, anodal tDCS impaired the set-shifting ability in Met-homozygotes but not in Val-carriers [89].

### 2.4. Age and Gender Effects

To conclude our revision of stable inter-individual factors influencing tDCS effects, we must consider demographic features such as gender and age. More than directly controlling tDCS effects, participants varying in gender and age present a series of differences able to modulate current effects (see Appendix A—section D for studies examples).

Previous evidence suggests that skull thickness varies with sex and age [90]. For instance, Russell and colleagues [91] found that males had a thicker spongy layer than females. Consequently, a higher current density reached the males’ cortex, with higher differences found when stimulating the temporoparietal regions than the frontal ones. Overlapping results have been reported by Bhattacharjee and colleagues [92] that simulated tDCS induced electrical fields on 240 individual MRIs representative of three age groups (young, middle age, and older adults). For the simulations performed over the younger individuals’ MRIs, a higher current density reached the male cortex for the parietal montage. Interestingly, the opposite pattern was found for the simulation in the older group of MRIs, with a higher current density reaching the females’ parietal and frontal regions. Across all age groups, CSF and gray matter volumes significantly predicted the current intensity estimated at the cortical target sites.

Hunold and colleagues [93] stressed the age-dependent effects of tDCS, focusing their research on children, adolescents, and young adults’ head models. They found higher current density on the gray matter surface in children than young adults, applying an intracephalic montage delivering 1 mA current intensity to the left primary motor cortex. Overlapping results have been found by Kessler and colleagues [94]. They suggested that, on average, children are exposed to higher electrical fields than adults when the same current intensity is delivered. Still, there is likely to be an overlap between adults with smaller head sizes and children.

Moreover, an age-specific influence of tDCS on cortical excitability of the primary motor cortex has been demonstrated. Indeed, Moliadze and her group [95] found that both 1 mA a- and c-tDCS resulted in a significant increase of MEP amplitudes lasting one hour after the end of stimulation while reducing the intensity at 0.5 mA cathodal tDCS decreased corticospinal excitability, but the 0.5 mA anodal tDCS did not result in any effect in adolescents and children. Conversely, older adults show a delayed response to 30 min of 1mA of anodal tDCS on the primary motor cortex [96]. Indeed, the most significant increase in corticospinal excitability occurred immediately after the stimulation in the younger group and 30 min later for the elderly. The authors speculated that the delay in the tDCS-induced plastic changes was due to the deterioration at the microstructural level involving glial cells in the aging brain.

It is worth noting that an increase in distance between the scalp and the cortex characterizes even healthy brain aging. Indeed, a widespread decline is observed in cortical thickness following an anterior-posterior gradient, with frontal and parietal regions more affected than temporal and occipital ones. This change is particularly relevant for males in the left hemisphere [97,98,99]. The CSF level increase due to cortical depletion has been reported to reduce the focality of the current [43].

In addition, there are strongly age-related modifications of the overall brain functional connectivity architecture: long-range connectivity decreases, whereas short-range connectivity increases [40,100]. Recently, some authors hypothesized that this reorganization could improve prefrontal network connectivity and thus facilitate the performance via the modification of NMDA/GABA receptors response, which is essential to promoting synaptic plasticity, otherwise reduced in the elderly. Indeed, Hanley and colleagues [101] found a significant improvement in task-switching speed after ten minutes of anodal tDCS compared to baseline in elderly adults.

Finally, gender differences play a role in the modulation of human cortical plasticity and excitability and thus in the effects induced by tDCS, possibly because of sex hormones and circadian cortisol blood levels [40,48,102,103,104,105,106]. For instance, Kuo and collaborators [107] found that the MEPs reduction induced by cathodal tDCS was prolonged in females compared to males. In contrast, the enhancement caused by anodal tDCS showed no gender difference.

## 3. Variable Individual Factors of Inter-Individual Variability

All the subjective characteristics described so far remain immutable for experimental sessions and constitute non-modifiable factors. However, other variables could change across sessions, making it even more complex to predict the electric field distribution in the brain and the behavioral effect within the same individuals across different stimulation sessions. For instance, Guerra and colleagues [40] identified several modifiable factors affecting brain excitability, such as medical and non-medical substance intake, quality of sleep and arousal, circadian and metabolic cycles, and cyclic hormone fluctuations.

Only a few studies investigated the effect of these factors on tDCS (see Appendix A), while previous works addressed this issue on another NIBS technique, namely TMS. Indeed, previous evidence suggested the influence of hormonal changes, neurotransmitter concentration, stress, age, and substance consumption on the resting motor threshold (RMT) [106]. RMT is an individual measure of corticospinal and—indirectly—cortical excitability. It is defined as the lowest stimulator output intensity at which pulses delivered over the motor cortex are enough to evoke a measurable muscular twitch [108]. RMT can then be considered a measure of stimulation sensitivity and is commonly used to establish the stimulation intensity when targeting cortical regions outside the primary motor cortex.

### 3.1. Endogenous Hormonal Variation

Serum cortisol exhibits circadian variations throughout the day, with a brief and variable rise in its levels in the morning and a progressive decline during the day [109]. The cortisol fluctuations affect brain function and neuroplasticity [48,110]. Indeed, Sale and colleagues [104] found increased MEPs after stimulation in the evening (when endogenous cortisol levels were low) but not in the morning. Intracortical inhibition is reduced during the day—and in general, with sleep deprivation—potentially affecting the mechanisms of neuronal plasticity. In repetitive TMS (rTMS) studies, higher cortisol levels predicted a more significant response to neuroplasticity protocol [23,111]. Moreover, depending on biological characteristics, the resting motor threshold was higher in elderly, stressed individuals and in women with amenorrhea [112].

Another variable factor influencing women participants is the menstrual cycle stage. Indeed, while estrogens enhance cortical excitability, progesterone decreases it [112]. Several studies tried to deepen its role with conflicting results. Some authors found differences in cortical excitability between the follicular and the luteal phases, increasing MEP through the follicular phase [102,113]. Moreover, MEPs were reduced in the females’ early follicular phase compared to men, while no gender differences were found in the late follicular phase [103]. Other studies found no significant changes in cortical excitation among the three stages [114,115] nor between females and males [116]. Translating results obtained with TMS to tDCS always requires some caution due to the different action mechanisms of the two techniques. However, the factors affecting TMS outcome measures, such as RMT or MEPs amplitude, might also critically impact the tDCS-induced effects. More research is needed to address how much these factors might contribute to the variability of the results at a cognitive-behavioral level, aside from neurophysiological ones.

### 3.2. Exogenous Non-Medical Substances Assumption

Brain excitability and the neurophysiological response to stimulation can also be altered by exogenous substance intake. Indeed, some previous studies suggested that glucose and caffeine might influence cortical excitability and cognitive functions [117]. Caffeine associated with anodal tDCS seemed to increase MEPs and muscular strength and lower the ratings of perceived exertion when associated with anodal tDCS [118].

Even nicotine can modulate tDCS effects [119,120]. For instance, the administration of a nicotine patch six hours before tDCS modulated the aftereffects in a dose-dependent way, abolishing the increment or reduction of excitability after anodal and cathodal stimulation, respectively. Nicotine consumption might affect neuroplasticity and improve cognitive performance mainly by increasing the calcium permeability and modulation of diverse transmitter systems. It has been observed that nicotine administration impairs calcium-dependent plasticity induced by tDCS in non-smoking participants, abolishing the anodal tDCS-induced neuroplasticity, probably because of intracellular calcium overflow [121,122]. So, we could expect a reduced effect of tDCS when including smokers among participants.

Lastly, even low doses of ethanol may have detrimental effects on LTP-like plasticity in the human motor cortex, affecting LTP-dependent processes, such as learning and memory formation [123].

Although relatively less common in the healthy population, other recreational substances should be considered. For instance, amphetamine consumption has been shown to prolong the excitability enhancing aftereffect under the anode and slightly reduce the excitability decrement typically seen under the cathode [124]. These issues undoubtedly become more critical when considering the potential use of tDCS in addiction treatment (for a recent review, see [125]).

### 3.3. Exogenous Medical Substances Consumption

A wide variety of medications can affect the effectiveness of tDCS stimulation (for a review, see [126]). Indeed, medication can alter the concentration of neurotransmitters, such as GABA, glutamate, dopamine, serotonin, norepinephrine, etc. This alteration might have a relevant impact on the mechanism underpinning tDCS online and offline effects.

Medications affecting either sodium or calcium channels, typically consumed by patients with cardiac disease, high blood pressure, neuropathic pain, and migraines, have been shown to impair the efficacy of tDCS. For instance, carbamazepine intake prevents the online and offline effects of anodal tDCS by blocking sodium channels [127,128]. Similarly, a medium or high dosage of dextromethorphan abolished any aftereffect induced by either anodal or cathodal stimulation. This effect is not surprising given that dextromethorphan is an NMDA antagonist and converging evidence suggests that LTP-like effects depend on NMDA receptors’ excitation [127,128,129,130].

The interaction between stimulation and dopamine medication is even more complex. This combination is modulated by both the dosage and the binding receptors. Administration of L-DOPA, a dopamine precursor for Parkinson’s Disease treatment, has an inverted-U dose-response effect on tDCS. Low (25 mg) and high (200 mg) dosages of L-DOPA eliminated the aftereffects of tDCS after both anodal and cathodal tDCS. In contrast, a medium dosage of L-DOPA (100 mg) switched the excitability enhancing aftereffects under the anode into excitability reduction and prolonged the excitability reducing aftereffects of tDCS for both anodal and cathodal stimulation [131].

Besides the drugs in use in clinical settings, several medications largely diffused among the “healthy population” might show interaction with the tDCS effect. Subjects with sleep disorder or anxiety may be treated via benzodiazepines that alter GABA concentrations and may impact the efficacy of tDCS [132,133]. For instance, Lorazepam, a GABA agonist, delayed but prolonged the excitability increment following anodal tDCS, with no effect after cathodal stimulation [132]. Similarly, selective serotonin reuptake inhibitors (SSRIs) altered the aftereffects of tDCS [134,135]. For instance, a single dose of Citalopram increased and prolonged the excitability-enhancing aftereffects of anodal stimulation. In contrast, in the case of cathodal stimulation, Citalopram reversed excitability reduction [134].

## 4. Contextual and Experimental Features as a Source of Inter-Individual Variability

Among variable subjective factors which might influence the effects of tDCS generating interindividual variability, several depend upon contextual and experimental features. The first crucial variable in the study design is whether the stimulation is coupled with task execution or delivered while participants are resting. Indeed, the state of the targeted areas during the stimulation has been shown to influence tDCS significantly—and generally NIBS—effects [136,137]. This phenomenon has been defined as *state dependency* [138]. According to this conceptualization, applying the stimulation concurrently with task performance may induce a synergistic relationship between an endogenous neural activity generated by the task’s execution and an exogenous source of activity caused by the tDCS [139,140,141,142]. This concept is also known as *functional targeting* [143], stressing the importance of matching the brain area to be stimulated with its specificity for the task that participants will perform (for a recent review, see [144]). In line with this idea, we ran a series of studies with transcranial magnetic stimulation co-registered with electroencephalography (TMS-EEG) recorded before and after tDCS combined or not with a task. We demonstrated that performing a task can change behavioral performance and modulate neuromodulatory patterns in the underneath functional cortical networks. For instance, anodal tDCS delivered during the resting state increased excitability in the Default Mode Network—a frontoparietal circuit activated when people are mind-wandering [5,6]. Conversely, applying anodal tDCS during a verbal fluency task triggered a neuromodulatory effect in areas involved in task performance, such as the left inferior frontal and the premotor cortices [7]. Still, it did not modulate the excitability of other regions not involved in task execution.

Interestingly, different results were found when cathodal tDCS was delivered at a resting state. Indeed, no changes were traceable at the electrophysiological level when comparing brain excitability before and after stimulation vs. sham conditions [8]. However, when the same stimulation parameters were applied during a visuospatial attentional task, a decrease in brain excitability in task-related regions combined with a reduction in task performance were found in the real compared to the sham condition [145]. This evidence may complement the observation in animal models that the neuromodulation exerted by tDCS occurs if neurons are activated during the stimulation protocol [84].

Given the state-dependent nature of aftereffects, some authors suggested that combining stimulation with cognitive treatments might improve behavioral outcomes and reduce inter-subject variability (see, for example, [146]). Despite largely agreeing with this idea, defining an unambiguous interaction between task engagement and stimulation effectiveness is challenging. Indeed, delivering stimulation concurrently with a task influences task performance and neuromodulatory patterns in the functionally connected networks performing the task. Crucially, task features and inter-individual differences—in terms of capacity and motivation—can influence the emergence of tDCS effects and participants’ engagement in the tasks. Differences at these levels can induce various amounts of neural activity that, in turn, might influence the magnitude of tDCS effects [147,148,149], thus generating inter-individual variability.

### 4.1. Baseline Capacity Levels as a Source of Inter-Individual Variability

If the presence of a task might influence the tDCS effects that much in terms of effectiveness and affected neural networks, it is intuitive that inter-individual differences in task performance can, in turn, become a critical source of variability. Converging evidence suggested that inter-individual baseline capacity in cognitive tasks might influence tDCS effects on behavioral performance (see Appendix A for examples of studies). In one of the first studies addressing this issue, Tseng and colleagues [150] engaged 20 participants in a visual working memory task after receiving 20 min of anodal or sham tDCS. No evidence of tDCS effects was traceable while analyzing data at the group level. Participants were then split into two groups of low and high performers based on their median score at sham task performance, which was considered a measure of pre-existing individual differences in capacity. Results suggested that anodal tDCS improved performance only in the low performers’ subset, while in the high performers’ subset, it did not. If anything, anodal tDCS in the high performers’ group seemed to worsen performance, even though such effect was only marginally significant. Subsequent studies explored the extent to which baseline capacity modulated tDCS effects on behavioral performance across several domains, like spatial attention [151], visuospatial working memory [152], and inhibitory control [153]. These works typically agree that tDCS effects at the behavioral level are influenced by individual capacity as measured in pre-stimulation or sham/placebo conditions, even though predicting which group or participants’ features will benefit from the stimulation remains controversial. Indeed, while several studies provided evidence supporting Tseng and colleagues’ [150] results [154,155,156,157,158], others highlighted different patterns. In some research, only good performers benefitted from the stimulation [159], while low performers had detrimental effects [151,152,160].

A few studies investigated whether behavioral differences in low vs. high performers were traceable also at a neurophysiological level (e.g., [150,161]). For example, Hsu et al. [161] measured electroencephalography (EEG) concurrently with a change detection task. They found more extensive changes in low vs. high performers after tDCS, mirroring their behavioral improvement. Indeed, pre-stimulation EEG signals in low performers showed high alpha amplitude in the parieto-occipital areas, an index associated with reduced attention/vigilance. After anodal tDCS, alpha activity decreased, and visual working memory performance improved. Conversely, high performers did not show a significant alpha activity at baseline, and tDCS did not affect its amplitude.

Different authors have interpreted this pattern as an effect of the state dependency of the stimulated functional network. State dependency is traceable for NIBS in general (e.g., [162,163]) but has been mainly documented in TMS studies (e.g., [164,165]). Several hypotheses have been accounted for explaining the phenomenon. A first interpretation could be that poor baseline performers have more room for improvement in measuring tDCS effects on behavior. Conversely, good performers could be nearer to the ceiling at baseline and therefore not benefit from neuromodulation. However, this interpretation does not entirely account for the results’ complexity, neurophysiological data, and reverse effects found for stimulation in good performers. Another, not mutually exclusive, hypothesis suggested that a poorer baseline performance would indicate lower brain excitability, whilst a better baseline performance would reflect higher excitability at baseline (e.g., [155,165]). Therefore, an increase in neural excitability induced by tDCS would facilitate performance in participants with initial low excitability. Conversely, tDCS would lead to the overactivation of networks in participants with high-initial excitability, thus impairing performance. In different terms, participants with low initial performance may be recruiting resources or pursuing strategies in a sub-optimal way at the baseline level and would benefit more from stimulation. In contrast, participants with high performance at baseline already have a good strategy in terms of cerebral circuit involvement. At this level, any exogenous stimulation would not improve the performance of this already-optimal process. The concept of stochastic resonance and optimal noise level have been proposed to formalize these hypotheses [138]. However, a crucial methodological issue concerns how baseline performance has been operationalized across the different studies (e.g., [166]). In some papers, authors referred to the baseline as the pre-stimulation assessment performance or the sham stimulation condition in crossover designs. In these experiments, participants can be split on a median performance level into low and high performers [151,167], or behavioral performance scores are correlated or used as regressors on task performance during or after the stimulation [168,169]). In other studies, the authors considered the performance at some measures representing general cognitive abilities as the baseline. For instance, Jones and Berryhill [159] split participants based upon a combined score obtained from digit span forward and backward, and tDCS effects were measured on performance in a change detection and a sequential presentation task. Crucially, as the study conducted by Hsu et al. [152] pointed out, different results can be achieved according to the different baseline measures selection. These authors ran analysis according to two methods for splitting participants (i) based on their pre-stimulation performance at digit span backward or (ii) at the Corsi tapping task, which was analyzed to measure tDCS effects. They found that anodal stimulation decreased performance in low initial performers, but only when split according to the Corsi performance at baseline. Conversely, no tDCS effects were found when participants were divided based on the digit span performance.

It is worth acknowledging that individuals’ beliefs and expectations may also play a role in the emergence of tDCS effects (see, for example, [170]). However, this issue has been explored in a recent work [171]; therefore, this point will not be discussed in the present review.

### 4.2. Task Difficulty as a Source of Inter-Individual Variability

Individuals’ baseline capacity is closely linked to another crucial issue to be considered when designing experiments, namely task difficulty. A few studies systematically investigated whether tDCS effects are differently traceable for different levels of tasks’ demands (for studies examples, see Appendix A), pointing out that they are not likely to emerge when task difficulty is low, probably due to a ceiling effect. For instance, Roe et al. [172] engaged participants in a multiple object tracking task, parametrically increasing the number of objects to be followed in each visual field, namely one (easy level), two (moderate level), or three (difficult level). Bihemispheric tDCS was applied during task execution over posterior parietal regions in three within-participants sessions: cathodal left/anodal right, anodal left/cathodal right, and sham. The authors reported that the two real stimulation conditions worsened performance compared to sham, but only at the higher attentional load. Pope and colleagues [149,173] engaged participants in a paced auditory serial addition task (PASAT) and a paced auditory serial subtraction task (PASST) performed before and after 20 min of anodal tDCS, cathodal tDCS, or sham condition had been delivered over the right cerebellum [173] or the lDLPFC [149]. The authors suggested that the two tasks had different difficulty levels, the PASST being more demanding. In both articles, the authors reported that no tDCS effects were traceable in the PASAT, while in the more difficult task, different patterns emerged, depending on the target region. Cathodal tDCS delivered over the right cerebellum improved accuracy compared to sham and anodal tDCS groups and reduced reaction times compared to the pre-stimulation session. In the second experiment, targeting the lDLPFC, they found an improvement in performance following anodal tDCS in the PASST.

Conversely, overdemanding tasks might prevent tDCS effects from emerging. For example, Kwon et al. [174] engaged participants in a visuomotor coordination task performed before and after anodal or sham tDCS. The difficulty was manipulated by changing task speed, implementing three difficulty levels: low, moderate, and high. Anodal tDCS improved performance, but only at the intermediate level, while no effects were traceable for the easier and the harder conditions. Why are tDCS emerging for some difficulty levels and not others? One of the possible explanations is relatively trivial. For difficulty levels that are too easy/hard, tDCS modulation might not emerge due to ceiling/floor effects that prevent the detection of improvements or worsening [150,175,176].

Moreover, individuals’ perception of task difficulty might influence whether they are likely to engage in the task or not. For example, motivational theories suggest that participants typically use just the resources necessary to succeed in a given job, but not more. Therefore, investing resources in a task perceived as impossible wastes energy, and participants disengage from its performance [177,178]. Another explanation comes from neuroimaging studies, which provided evidence pointing out a well-established correlation between task demand increase and brain overactivation [179,180,181], suggesting that neural regions are “working harder” to face increasing task demands. However, this increment is linear until it reaches a resource ceiling (see [182]), followed by a decrease in brain activation and performance, showing a typical inverted-U curve [183,184]. In other words, when a task is easy, it will require a low level of activation that will grow with the task’s demand until it reaches a resource ceiling, which is different from one individual to another. Once the individual resource ceiling is reached, the activation will decrease even if task difficulty increases, and behavioral performance will worsen. These neuroimaging findings are crucial if we consider how tDCS can influence neural activity. Indeed, it is well-known that the current injected by tDCS is too low to induce action potentials directly. Instead, the current can modulate excitability at the membrane potential level, thus modifying the neural firing likelihood by causing subthreshold changes. It seems reasonable that these changes would induce more significant effects in neural regions that are already activated because they are engaged in a task than networks at rest [7]. Thus, coupling tDCS with a specific concurrent activity involving the same target region can boost behavioral stimulation effects AND spatial resolution of a technique that is otherwise thought to be low [53].

Considering how task difficulty impacts inter-individual differences, a crucial point concerns how this concept has been considered in the literature. Indeed, the lack of a shared definition of task difficulty led to an inevitable confusion in how difficulty has been manipulated across studies (see for a revision [185]. In some of the reviewed studies, the level of difficulty has been manipulated by increasing only one parameter, for example, the number of competing stimuli to be detected [186,187], of items to be remembered [188], the length or syntactic complexity of sentences [147,189,190], objects’ conventional vs. unconventional view [191]. These manipulations, however, can be misleading. First, participants’ strategies to face different versions of a task can change at a quantitative and qualitative level—for example, in the shift from 1-back to 2-back/3-back tasks [192,193,194]—that can involve different neural substrates. Secondly, people can have differing opinions of what is difficult and what is not, and using priori established levels of difficulty, totally ignore these differences.

## 5. Discussion

In the present review, several factors contributing to the inter-individual variability have been explored and discussed, providing evidence, open questions, and limitations of the available literature. The first part of the revision focused on the variability concerning the individual characteristics. Following earlier works [40], we classified them in stable vs. variable factors, including morphological, genetic, and demographical characteristics vs. circadian, hormonal, and substance-intake variations. In the second part, in a more functional perspective, we explored another variable factor of inter-individual differences concerning experimental features and specifically how individuals’ baseline capacity and task engagement can influence tDCS effects measured at a behavioral and neurophysiological level.

From our perspective, the inter- and intra-individual variability in tDCS effects undermines the reliability and, in the end, the technique’s effectiveness. However, rather than neglecting this issue or jumping to a drastic conclusion on tDCS’s ineffectiveness, we believe that deepening the knowledge of the factors inducing such inter-subject variability might be beneficial to understanding how stimulation protocols can be optimized. We then suggest some good practice norms that can be considered in the study design, implementation, and analysis. These norms are not meant to be comprehensive; instead, they focus on specific parameters and procedures that could be useful to minimize potential factors inducing individuals’ differences.

Of course, a preliminary suggestion is choosing appropriate sample sizes. TDCS studies tend to be underpowered, i.e., with low statistical power due to insufficient sample size, thus dampening results’ reliability and reproducibility. According to a recent meta-analysis, the average sample size for within-subjects studies is *n* = 18, with a statistical power ranging only from 8 to 16% [195]. However, the poor statistical power is a widespread concern in NIBS and cognition literature and represents one of the main reasons for inconclusive findings in several domains. Hints regarding sample size are then proper for any study design, independently from tDCS application. In this discussion, we instead focused on concerns specific to tDCS employment.

### 5.1. Data Analysis and Results Presentation

Starting from the idea that inter-individual variability should be acknowledged, a good practice implies a change in how the data are analyzed and the results reported. We recommend showing individuals’ variability and considering implementing statistical approaches to account for individual differences. In contrast to the typical average-based ANOVA approach, statistical analyses that better take the variability of outcome measures are recommended—as mixed models analyses [196,197,198]. These analyses can assess fixed factors manipulated by the experimental design and parameters that cannot be controlled but can impact results if not correctly considered. By-participants’ random structures can be included in statistical models to account for the non-independency of observations in repeated measurements experiments and address inter-individual differences better.

Moreover, a parallel individual-level presentation should be implemented aside from a group-level description. Rather than answering whether *tDCS is effective or not*, in which the answer is fully binary, the right question to be tested would become *how much is tDCS effective?* i.e., *how many participants are affected?* This change would reveal—rather than hide—the interindividual variability in response to tDCS, shifting the attention to the participants’ or experiments’ characteristics influencing the modulation. Furthermore, a more thorough analysis of the factors determining whether stimulation was effective or not can increase our knowledge of the basics of tDCS mechanisms of action and experimental design. As discussed in the introduction and shown in Appendix A, many studies suggested that experimental samples have different percentages of responders and non-responders. Provided an adequate baseline/sham control condition, responders are those whose tDCS effects, i.e., the performance in each task, are significantly different in the real vs. control condition.

When high variability is present, the main effect at a group level might cover the presence of subgroups in which the same effects are not visible or even reversed. As a title of example, Figure 1 shows the distribution of results in the four TMS-EEG studies performed by our research group previously described, three of which have already been published [5,7,8]. In three out of four experiments (anodal at rest [5]; anodal with the task, [7]; cathodal with the task, [145]), significant differences in the sham vs. real stimulation conditions were reported at the group level. However, looking at the inter-individual variability, a remarkable rate of non-responders is present. In the same vein, in Varoli et al. [8] (cathodal at rest), no differences were reported at the group level, even though some participants seemed to show an effect of real stimulation.

The way responders and non-responders have been defined is a central problem. Indeed, the division is typically based on arbitrary criteria, such as a >50% pre-post change, median split, etc. A recommendable statistical approach to identify naturally occurring subgroups that may be present in the whole sample is cluster analysis. Cluster analysis is an analytical data-driven subgrouping method that separates data based on patterns or trends ([199]; see [28] for a recent revision on cluster analysis applied to NIBS). However, this approach is feasible only for large samples. A graphical presentation of results can be of help in summarizing data distribution. Indeed, many studies (and ours among them) use bar and line graphs to show findings, typically presenting mean/median values and standard errors/deviations. However, the use of scatterplots (as in Figure 1A), dot plots, etc., allows readers (and authors) to have a clearer picture of data distribution [200,201].

The different methods (responders vs. non-responders analyses, cluster analyses, data presentation) entail a reconsideration of null results at a group level. Indeed, in the presence of high inter-individual variability, caution is needed in interpreting null results at the group level. If the sample size and study design are appropriate, null results might hide an internal mixed pattern of results, where aside from many non-responders, a smaller group of responders might be present. In this vein, a Bayesian approach to the amount of information present in the dataset, both towards a null or alternative hypothesis significance, can help disentangle the issue.

Moreover, when the tDCS aftereffect time course is investigated, separate cluster analyses at each time point might unveil interindividual differences in the onset and duration of tDCS effects. For instance, the study by Luque-Casado et al. [29] explored the effects of anodal vs. sham tDCS over the left dorsolateral prefrontal cortex (DLPFC) during the performance of a digit span backward task. Memory span performance was assessed before (baseline), immediately after tDCS administration (T1), and 10 min post-T1 (T2) a single session of either anodal or sham tDCS. The results at a group level showed that anodal offline tDCS failed to improve the performance in the working memory task in the whole sample. The following cluster analysis identified the presence of a subgroup of responders that significantly improved their performance after anodal (vs. sham) tDCS in T1 (47%) and T2 (46%). Crucially, only seven of these responders showed a consistent improvement in performance for both T1 and T2. In contrast, five of the responders in T1 failed to maintain their improvement in T2, and six ‘responders’ increased their performance only in T2 (but not in T1). These findings support the time dependency of tDCS induced effects and highlight a certain degree of interindividual variability in the tDCS aftereffects time course. Therefore, in offline tDCS designs, whenever possible, planning longitudinal time points for outcome assessment would better account for this variance.

### 5.2. Computational Modeling to Set tDCS Parameters

As described before, variance in scalp and brain morphology (e.g., gyrification, skull-to-scalp distance) crucially affects the amount and distribution of tDCS-induced current in the brain.

Since the early days, finite element head models have been implemented to investigate the magnitude, spatial distribution, and direction of currents delivered over the scalp and reaching the brain. The first models were relatively simple but still allowed to determine some principles that are still solid today. Results suggested that approximately half of the current delivered by the tDCS was shunted through the scalp [52]. Crucially, increasing the distance between the electrodes decreased the current shunted through the scalp, growing, in turn, the current reaching the brain. A limitation of this model is that it did not account for other essential layers, representing, for example, the CSF and the *dura mater*, the complex composition of the skull itself, and the geometry of brain tissue that affect the electric field and current density reaching the brain (for a review, [43,202,203]). More recently, Opitz and colleagues [43] analyzed the electric field induced in 28 different electrode montages obtained moving the anode in steps of 5 mm (with anterior-posterior, medial-lateral, and rotation movement) applied to an anatomical model of two healthy subjects’ heads. For each head model, they computed skull thickness, CSF thickness, sulcal depth, and gray matter’s distance to the electrode edges of the anode. All these factors accounted for up to 50% of the spatial variation of the electric field strength that reaches the targeted cortical area. Indeed, they confirmed that the field strength was higher (i) where the skull was thinner or is thicker but composed of spongy bone; (ii) where there was a thick CSF layer—being the CSF the tissue with the highest conductivity in the brain; (iii) at more superficial brain regions and to a less extent within the sulci; and (iv) near to the electrode edges and dependent on the average distance to the reference electrode.

As the authors suggested, individual differences in anatomical features could influence the current distribution, for example, generating *hotspots* that are partly resistant to electrode positioning.

Nowadays, more accurate 6-layer models propose considering the scalp, skull differentiated in spongy and compact bone, CSF, gray matter, and white matter [45].

Whereas some computational modeling approaches can hardly be implemented because they require high-definition individual MRI and time-consuming computations, there is freely available and accessible software to apply, such as COMETS ([204], http://www.COMETStool.com) or SimNIBS ([205], https://simnibs.github.io/simnibs). In this perspective, the good practice would be to use such computational modeling software to estimate the amount and distribution of induced electrical fields for a given montage or intensity. This approach would allow controlling the appropriateness of the chosen current intensity and the electrodes montage in advance or personalization of the electrode montage and the stimulation parameters to target a specific cortical area [44,71,206]. However, further data are needed to confirm the effectiveness of these modeling to define the target location (e.g., tDCS during fMRI).

A step further would be to use such modeling to select intensity and electrodes montage individually. Both elements affect the electric field reaching the brain and behavioral outcome (see Appendix A for some examples). The use of individual MRI might be relevant when dealing with brain-damaged patients, where the presence of brain lesions might significantly influence the way the induced current is shaped.

Although an individualized montage seems effective in dealing with inter-individual anatomical differences, it could be speculated that it increases variability in the spatial distribution of current flows across brain regions of different subjects. Indeed, anode and cathode positions strongly determine the electric field distribution, with the electric field being stronger near the anode’s edges. In contrast, a combination of the “local” distance to the anode edges and the more “global” distance to the center of the cathode accounts for the electric field distribution in the cortex positions in its proximity [49,207]. So, to overcome this issue, a possible suggestion would be to maintain fixed electrode placement and change tDCS intensity across different participants to account for inter-individual morphological differences [206].

For other non-invasive brain stimulation techniques, such as transcranial magnetic stimulation (TMS) TMS, the stimulation intensity is typically individually set as a percentage of the RMT. Conversely, for tDCS, the conventional application employs a fixed current intensity across the entire experimental sample. Most of the studies set tDCS at an intensity of 1 or 2 mA without knowing whether this current is enough at the participant level to modulate brain excitability [27,42,48,202,206,208]. Dmochowski and colleagues [209], for instance, found a 64% increase in electric current strength in the targeted cortical area and 38% improvement in a behavioral task using individualized tDCS montage in stroke patients using high definition tDCS.

Together with stimulation duration, current density has been shown to determine the enhancement of anodal tDCS on the excitability of the motor cortex and motor function in healthy individuals and subjects with stroke in a review by Bastani & Jaberzadeh [210].

Several studies highlighted that each individual has an optimal threshold that must be reached with adequate current intensity [27,211,212]. Moreover, these studies demonstrate reliability between subsequent repeats of the same protocol [101]. Indeed, while increased intracellular calcium induces long-term potential, exceeding optimal levels will activate potassium channels and induce hyperpolarization [213]. This change could account for the suppression of expected neuromodulatory effects (e.g., [214]). Generally, lower doses—i.e., 1 or 2 mA—seem enough to obtain the desired selectivity in younger adults, while higher intensities—like 3 mA—result in better selectivity in stimulating the targeted brain area than the non-target region in older adults, especially males [215]. Interestingly, studies have shown that higher doses of up to 4 mA could be safely used, even if it is less tolerable in females than in males [216,217]. Furthermore, typical inhibitory effects of cathodal stimulation could disappear at high intensities [27] or be reversed, generating excitation due to excessive stimulation and habituation of potassium channel response. This modulation could determine an enhancement of performance following inhibitory stimulation [101,211,218].

Individuals’ neurobiological circuits have homeostatic constraints to prevent over-excitation of calcium channels and NMDA (N-methyl-D-aspartate) receptors, and this effect is readily observed at various intensities [81,101,219,220]. This evidence suggests that a specific cohort might need higher doses while low intensities might be sufficient for others.

Higher current strengths are not always necessary to produce modulations of excitability. At the same time, stimulation that is insufficient to fulfill an individual’s optimal threshold may propagate deficient calcium transmission. Investigating the stimulation intensity-dependent effects of tDCS on motor cortex excitability in healthy subjects, Batsikadze and colleagues [211] found a non-linear effect. Indeed, at 1 mA, cathodal tDCS decreased corticospinal excitability, while at 2 mA anodal and cathodal tDCS induced MEPs amplitude to increase. Similar results were found by Wiethoff and colleagues [27], whose three-quarters of participants showed facilitation after 10 min of anodal tDCS on the motor cortex and one-quarter inhibition. In contrast, after cathodal tDCS, the proportions were approximately 60:40 (facilitation: inhibition).

Non-linearity has also been found outside the motor regions. For example, Nikolin and colleagues [221] applied bihemispheric tDCS over the dorsolateral regions (anode over the left, cathode over the right) during a visual 3-back working memory task. Participants received 15 min of bihemispheric frontal stimulation at different current intensities (1 mA, 2 mA, and three different sham conditions: 0.034, 0.016, 0 mA). Although behavioral effects did not differ in the stimulation condition, EEG activity suggested a difference between the 0 mA sham condition compared to the 1, 2, and 0.034 mA conditions, with the more significant effect sizes reported for the 1 mA condition.

To sum up, the use of computational modeling software is recommended to estimate, with different degrees of precision according to their complexity, the tDCS induced current flow. Their use can optimize tDCS parameters by guiding electrodes montage or adjusting the intensity across individuals, hence better accounting for inter-individual variability. However, two crucial points should be discussed considering electric field modeling. First, they are only an estimate of the current distribution in the brain. Indeed, many studies with in vivo or in vitro direct measurement of DC effects were performed on animals, but typically, intensities are not comparable to those applied to humans. Only a few studies [222,223,224,225,226] measured in-vivo electric fields in humans, since invasive methodologies are required. In vivo human studies have been performed only in patients with implanted depth electrodes, typically due to drug-resistant epilepsy or movements disorders such as Parkinson’s disease. These studies combined intracerebral recordings during stimulation delivered through tDCS [224] or tACS [222,223,225,226] (see Appendix A for studies features and results description). The studies have the great value of demonstrating that transcranial electric stimulation delivered over the scalp can also reach significantly deep brain structures. Moreover, some of these studies compared estimated electric fields provided by modeling with the measured electric fields [223,226], providing empirical support to the models’ accuracy. Secondly, future research should address the effect of electric fields’ strength on the behavioral outcome to clarify how morphological differences can translate into different tDCS-induced neurophysiological and behavioral effects.

### 5.3. Task and Outcome Measure Selection

Task choice is a critical point of our procedure for building experiments. When we choose a proper task, we must keep in mind some key points: (i) the task should be selective in measuring the function we are targeting; (ii) the task should be sensitive to allow for the detection of small changes, (iii) the task difficulty should be challenging for participants, avoiding ceiling/floor effects, and motivating them to perform the task (see also [227]). It is essential to minimize differences as much as possible at this level by individually settling a similar level of difficulty that considers individuals’ differences. In previous studies, individuals’ levels of difficulty have been set by choosing a certain percentage of correct responses before starting the stimulation protocol allowing some range of improvement and impairment (e.g., the level at which participants perform the 80% of trials correctly [38] or establishing individual thresholds [185]. It is essential to remember that some measures are more stable than others are and will be more difficult to modulate. For example, many studies investigated the possibility of changing digit span forward capacity using tDCS, which is quite a stable measure ending in unsuccessful trials.

Concerning the behavioral outcome measure, previous studies suggested that tDCS seems to be more effective in inducing changes in reaction times than accuracy [18,25]. This result aligns with what we stated above: reaction times are a more sensitive measure than accuracy, especially in healthy participants. Therefore, we suggest recording more than one outcome measure with different sensitivity levels.

Task selection is closely related to other issues concerning the experimental design, such as the target region to be stimulated, the time at which delivering tDCS compared with task performance (i.e., online vs. offline paradigms), and the presence of a proper control condition.

Regarding tDCS timing, as Appendix A pointed out, according to some authors (see for a recent review [146]), time-locking behavioral tasks with stimulation may improve outcomes and reduce inter-individual variability in response to stimulation. Several pieces of evidence from different research approaches converge in showing the state dependency of tDCS modulation. By combining in vitro and computational modeling, Lafon and colleagues [228] considered a possible explanation of the role of location and frequency of the active synapses, hence considering tDCS neurophysiological modulation as strictly connected to spontaneous firing and synaptic efficiency. Accordingly, in the animal model study by Fritsch et al. [84], M1 mouse slices needed simultaneous DC and synaptic activation to induce LTP-like changes. Therefore, it is presumed that the concurrent area activation through a task execution determines long-lasting plastic changes [84]. As previously reported, the results of TMS-EEG studies of our research group converged by showing that delivering tDCS during task execution limited the modulation of cortical excitability to the active functional network and entailed the occurrence of a neurophysiological modulation after cathodal stimulation.

We suggest that time-locking stimulation with a concurrent task can be the first step in decreasing heterogeneity, although it may not be sufficient. Moreover, future studies involving NIBS should be conducted at a fixed time of day, preferably in the afternoon, to maximize neuroplasticity and reduce variability [105]. Crucially, to investigate the influence of inter-individual differences in tDCS effects, authors should perform studies in which the sample is homogeneous in stable factors. In contrast, the experimenters should systematically manipulate one variable factor at a time to deepen the knowledge of the influence of each element.

## 6. Conclusions

In this narrative review, we covered relevant inter-individual factors potentially contributing to tDCS variability at the intra-study level. We acknowledge that all these sources of variability represent a limitation to the extensive use of tDCS, which is even more prominent if we decide not to consider differences as a matter of fact. The issue of inter-subject variability is not confined to tDCS but is also relevant to other NIBS techniques (concerning TMS, see [229], tACS [230]). More generally speaking, individuals’ differences are present in all treatments, from drugs to psychotherapy, but no clinician would say they are ineffective treatments. Therefore, after presenting evidence describing inter-individual difference sources, we presented several solutions to consider and address individual differences in our experimental designs, data collection, analysis, and presentation. These guidelines do not pretend to be exhaustive; however, we hope that they will be useful to trace a state-of-the-art ad to open new research avenues, shifting from *whether* tDCS is effective in modulating behavior to *how* it can be effective.

To conclude, we acknowledge that choosing a narrative review instead of a systematic literature revision is a limitation of the present paper. However, given the topic’s depth and complexity, we do not pretend to cover all the possible works. Instead, we hope to raise issues that increase the discussion concerning individual variability in NIBS studies.

## Figures and Tables

**Figure 1 brainsci-12-00522-f001:**
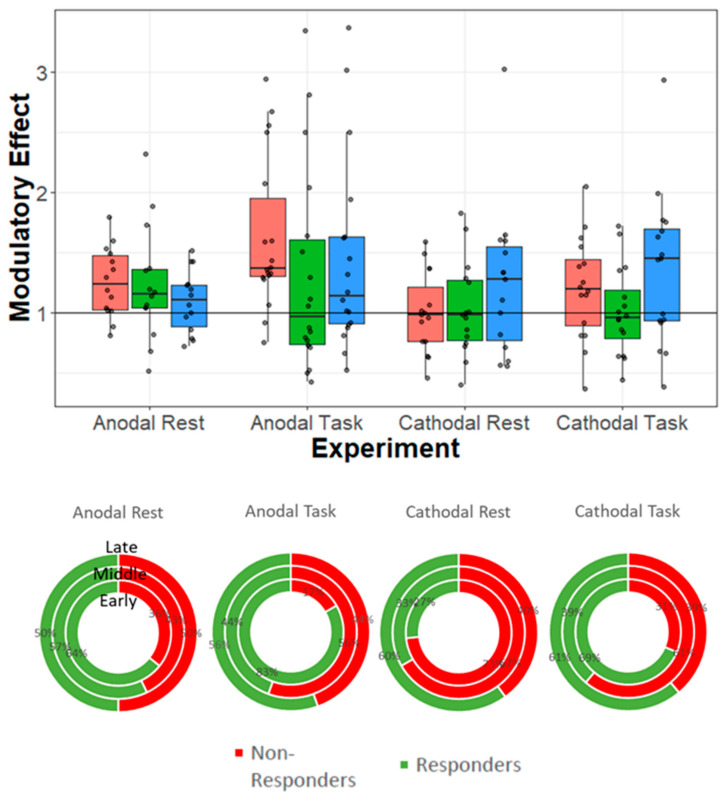
Patterns of responders and non-responders across the four TMS-EEG studies from our lab. The upper panel represents the interindividual variability of the modulatory effects of tDCS in early (0–50 ms), middle (50–100 ms), and late (100–150 ms) latency TEPs components, represented by the pink, green, and blue boxes, respectively. The effect was computed by dividing post-tDCS by pre-tDCS local mean-field power values so that values above 1 represent an increase in post-tDCS response. Cathodal results are reversed for graphical reasons. The bottom panel represents the percentage of responders (green) and non-responders (red) calculated as participants showing the outcome modulation of at least 20% of the pre-tDCS values on early, middle, and late TEPs components, represented by the inner, intermediate, and outer circles.

## Data Availability

Not applicable.

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
