# Peer review of "Inter-Individual Variability in tDCS Effects: A Narrative Review on the Contribution of Stable, Variable, and Contextual Factors"

_brainsci, 2022, doi:10.3390/brainsci12050522_

Round 1
Reviewer 1 Report
Overall, the topic of the study is important and aim to cover different factors contributing to the variability of the tES effects we all see in the literature. Despite several important points which I believe should be put in the spotlight (I personally agree with the relevance of all the issues raised in the paper), I see several critical issues with the paper as it is right now. Namely authors describe different studies on both healthy and clinical population with outcome measures from motor to cognitive performance all packed together. As a result, the text appears poorly structured as the information is scattered and not comprehensively presented. I understand the need to cover different fields, but the mechanisms and the effects may be significantly different for clinical and non-clinical samples. Therefore, I believe authors should either restructure the information or narrow down their focus.
The main issue is that it remains unclear to me how did the authors select the papers to be included in the review. without explicit and transparent literature search strategy many relevant works are easily omitted. As a result - the conclusions based on selected literature may be misleading or simply not comprehensive enough. For example, I know few papers that might be relevant in this context, but I don’t see them included. Also, some of my own work deals with the issues raised here but they are not included. This makes me wonder if and how many other works that I am not aware of have been omitted too, and if the conclusions would be the same or different if those works were included. I believe authors need to conduct comprehensive literature search and report on the search strategy as well as how the studies were selected or excluded to make this paper and its content credible. In the current format e.g. example of the studies tables, the paper could be criticized for biased selection of papers to be included in the review, as it does not adhere to the review standards. I don’t believe asking for a few references to be added solves this issue. I would advise authors to consult PRISMA guidelines – it does not have to be a comprehensive review abut it has to have explicit search and selection criteria for inclusion of the studies.
Author Response
Reviewer 1
Overall, the topic of the study is important and aim to cover different factors contributing to the variability of the tES effects we all see in the literature.
We thank the reviewer for her / his appreciation and the inputs for improving our work.
Despite several important points which I believe should be put in the spotlight (I personally agree with the relevance of all the issues raised in the paper), I see several critical issues with the paper as it is right now. Namely authors describe different studies on both healthy and clinical population with outcome measures from motor to cognitive performance all packed together. As a result, the text appears poorly structured as the information is scattered and not comprehensively presented. I understand the need to cover different fields, but the mechanisms and the effects may be significantly different for clinical and non-clinical samples. Therefore, I believe authors should either restructure the information or narrow down their focus.
We thank the reviewer for her / his comment. We agree that we raised different points in the manuscript. However, the present work discusses the evidence supporting the idea that inter-individual differences are common in tDCS practice and should be acknowledged within the field rather than shoved under the rug. The problem is intrinsically present in any data collection, no matter how controlled the sample, paradigm, and data collection are (even though, of course, attention to these points may reduce the variability). We agree that brain stimulation may differ when delivered over primary sensorimotor regions vs. associative networks. Still, as previous evidence highlights (see, for example, Table 1S), the examined factors inducing interindividual differences seem critical regardless of the targeted areas and tapped functions.
We agree that referring to clinical and non-clinical samples could instead be confounding. Therefore, we mostly narrowed the focus on healthy samples, keeping the references to clinical studies only when the impact of dealing with a clinical sample renders the examined factor more prominent, as in the case of the presence of brain lesions which might critic impact the current distribution or the contribution of exogenous substance intake in the case of Substance Dependence disorder. All the other references to the clinical samples have been removed.
The main issue is that it remains unclear to me how did the authors select the papers to be included in the review. without explicit and transparent literature search strategy many relevant works are easily omitted. As a result - the conclusions based on selected literature may be misleading or simply not comprehensive enough. For example, I know few papers that might be relevant in this context, but I don’t see them included. Also, some of my own work deals with the issues raised here but they are not included. This makes me wonder if and how many other works that I am not aware of have been omitted too, and if the conclusions would be the same or different if those works were included. I believe authors need to conduct comprehensive literature search and report on the search strategy as well as how the studies were selected or excluded to make this paper and its content credible. In the current format e.g. example of the studies tables, the paper could be criticized for biased selection of papers to be included in the review, as it does not adhere to the review standards. I don’t believe asking for a few references to be added solves this issue. I would advise authors to consult PRISMA guidelines – it does not have to be a comprehensive review abut it has to have explicit search and selection criteria for inclusion of the studies.
We thank the reviewer for this consideration. As mentioned through the manuscript, the present work is a narrative review. The articles included in the tables are representative examples and do not pretend to cover the available literature exhaustively. The narrative reviews are not required to be based on PRISMA guidelines, which are necessary for systematic reviews and meta-analysis (see, for example, Vergallito et al., 2021). However, for the sake of clarity, in the revised version, we highlight that this work is a narrative review by changing the title from “The issue of inter-individual variability in tDCS effects: the contribution of stable, variable, and contextual factors” to the revised version “Inter-individual variability in tDCS effects: a narrative review on the contribution of stable, variable, and contextual factors”. When referring to the tables (moved to the supplementary materials as suggested by Reviewer 2), we now specify that they provide “studies examples” and not “summaries”).
Moreover, we added this point as a limitation in the manuscript conclusion “To conclude, we acknowledge that choosing a narrative review instead of a systematic literature revision is a limitation of the present paper. However, given topic depth and complexity, we do not pretend to cover all the possible works. Instead, we hope to raise issues that increase the discussion concerning individual variability in NIBS studies.”
We acknowledge that this decision has the limitation of missing relevant issues and works and we would be happy to consider any proposal the reviewer would like to suggest.
Reviewer 2 Report
I found this to be a well-written, thoughtful, and informative review. There is little for me to critique. I have written some relatively minor suggestions I hope contribute to improving the manuscript.
My sense is that the Tables presented in the body of the manuscript belong in the Supplementary material. The information there, though comprehensive, cannot be digested easily or at a glance (neither of which is necessary because the information is already summarised as part of the review).
“Indeed, anodal tDCS resulted detrimental in COMT Met/Met homozygous individuals [92].”
The sentence phrasing is confusing here. Is it intended to state that tDCS had detrimental effects in COMT Met/Met individuals?
“Recent modeling studies, for example, suggested that the resting motor threshold could be influenced by neurotransmitter concentration, stress, age, sex, and phase within the menstrual cycle [109].”
Although there is a significant degree of overlap between those interested in tDCS and TMS, given both are forms of non-invasive brain stimulation, it should not be assumed that an individual interested in reading about factors contributing to tDCS variability would immediately understand the significant of a resting motor threshold (i.e., as a means to assess an individual’s sensitivity to TMS). This sentence should either introduce the concept of RMT, be rephrased using less jargon (e.g., by saying that ‘sensitivity to TMS can be influenced by neurotransmitter concentration…’), or omitted.
“Caffeine associated with anodal tDCS was proven to increase MEPs and muscular strength and lower the ratings of perceived exertion when associated with anodal tDCS [126].”
The language here is overly strong. It is difficult to conceive how a single study with a sample of 15 individuals could ‘prove’ an effect. This level of confidence requires several RCTs, collectively reporting on findings from hundreds or thousands of participants.
“…with athe statistical power ranging only from 8 to 16…”
Is this suggesting that statistical power is at 8 to 16%? Statistical power is typically expressed as 1-beta, which can be expressed as a fraction (0 to 1) or as a %.
5.1.Data analysis and results presentation
A point which is made implicitly with the display of Figure 1A, but should be made explicit, is the value of presenting data from all participants as a scatterplot. This allows a quick visual representation of inter-individual variability (see https://doi.org/10.1371/journal.pbio.1002128).
“In this sense, a good practice would be to always report the percentage of responders/not-responders.”
I’m in two minds on this point. On the one hand, I agree that this additional information may help contextualise some of the variability in tDCS effects. However, simple reporting of responders/non-responders fails to recognise the variability inherent in measuring a value from a participant for an outcome of interest. Using the criterion that a >50% change equates to being a ‘responder’, is a participant whose performance increases from 100 to 151 (a change of 51%) a true example of a ‘responder’ or at best a partial responder with the benefit of a positive contribution from measurement error? This issue remains true regardless of the criterion used to define ‘response’. Resolving the issue using a cluster analysis is only feasible for large samples, otherwise the solutions to these analyses are close to arbitrary median splits.
“Investigating the stimulation intensity-dependent effects of tDCS on motor cortex excitability in healthy subjects, Batsikadze and colleagues [222] found a non-linear effect.”
This has also been hinted at in healthy subjects with prefrontal stimulation, showing effect sizes that are greater at 1mA than 2mA (https://doi.org/10.1016/j.brs.2018.01.003)
“Regarding tDCS timing, as the 4 and 5 pointed out, according…”
It’s unclear to me who the ‘4’ and ‘5’ are? Is this a citation?
Minor text issues
“However, opposite conclusions were offered by Unal and colleagues [66], how failed to find a correlation…”
Should read ‘who’
“It is possible that nicotine assumption might affect neuroplasticity and improve cognitive performance…”
I believe this should state ‘nicotine consumption’.
“…typically assumed by patients with cardiac disease or high blood pressure…”
As above, this should state ‘typically consumed’.
“Similarly, Serotonin Selective serotonin reuptake inhibitors (SSRIs) altered the after-effects of tDCS…”
First instance of ‘serotonin’ should be deleted.
“4.Contextual and Experimental features as a source of inter-individual variability”
The formatting for this section is in italics, unlike the remainder of the manuscript body.
“Converging evidence indeded suggested…”
Typo
“Of course, a preliminar suggestion to be made regards the appropriate sample size and selection.”
Typo – should state ‘preliminary’
Author Response
I found this to be a well-written, thoughtful, and informative review. There is little for me to critique. I have written some relatively minor suggestions I hope contribute to improving the manuscript.
We thank the reviewer for her/his appreciation and the valuable suggestions to improve our work.
My sense is that the Tables presented in the body of the manuscript belong in the Supplementary material. The information there, though comprehensive, cannot be digested easily or at a glance (neither of which is necessary because the information is already summarised as part of the review).
We agree that tables were quite abundant. Therefore, as suggested by the reviewer, we moved them from the main text to the Supplementary Materials. In the first part of the manuscript (i.e., the discussion of anatomical, morphological, and variable factors related to hormones, neurotransmitters, and drugs), the studies included in the tables were already cited in the main text; therefore, we did not implement new references. However, in the section discussing factors related to the task, most evidence was cited only in the tables. Therefore, in this part, citations have been added to the main text to emphasize the previous literature properly.
“Indeed, anodal tDCS resulted detrimental in COMT Met/Met homozygous individuals [92].”
The sentence phrasing is confusing here. Is it intended to state that tDCS had detrimental effects in COMT Met/Met individuals?
We rephrased the sentence as follows “The COMT Val/Met polymorphism modulates the impact of tDCS applied to the left dorsolateral prefrontal cortex during the administration of tasks based on executive function, as the Go/No-Go paradigm. Indeed, cathodal tDCS impaired response inhibition in COMT Val/Val homozygous but not in Met-carriers [87]. In another work, anodal tDCS impaired the set-shifting ability in Met-homozygotes but not in Val-carriers [88].”
Moreover, we eliminated the following lines “The authors hypothesize that anodal tDCS further increased the elevated levels of frontal dopamine in Met carriers, thus exceeding the optimal levels and resulting in an inhibitory effect, according to the “inverted-U” theory of cognition [89].” Indeed, this was a speculative hypothesis made in the original paper.
“Recent modeling studies, for example, suggested that the resting motor threshold could be influenced by neurotransmitter concentration, stress, age, sex, and phase within the menstrual cycle [109].”
Although there is a significant degree of overlap between those interested in tDCS and TMS, given both are forms of non-invasive brain stimulation, it should not be assumed that an individual interested in reading about factors contributing to tDCS variability would immediately understand the significant of a resting motor threshold (i.e., as a means to assess an individual’s sensitivity to TMS). This sentence should either introduce the concept of RMT, be rephrased using less jargon (e.g., by saying that ‘sensitivity to TMS can be influenced by neurotransmitter concentration…’), or omitted.
Following the reviewer’s suggestion, we opted for introducing and clarifying the concept of the resting motor threshold. The following passage has been added “Indeed, several pieces of evidence suggested the influence of hormonal changes, neurotransmitter concentration, stress, age, and substance consumption on the resting motor threshold (RMT) [99]. RMT is an individual measure of corticospinal and – indirectly – cortical excitability. It is defined as the lowest stimulator output intensity at which pulses delivered over the motor cortex are enough to evoke a measurable muscular twitch. RMT can be considered a measure of stimulation sensitivity and is commonly used to establish the stimulation intensity when targeting cortical regions outside the primary motor cortex.
“Caffeine associated with anodal tDCS was proven to increase MEPs and muscular strength and lower the ratings of perceived exertion when associated with anodal tDCS [126].”
The language here is overly strong. It is difficult to conceive how a single study with a sample of 15 individuals could ‘prove’ an effect. This level of confidence requires several RCTs, collectively reporting on findings from hundreds or thousands of participants.
We agree with the reviewer’s suggestion and rephrased this passage as follows: “Exogenous substances intakes can also alter brain excitability and the neurophysiological response to stimulation. Indeed, some previous studies suggested that glucose and caffeine might influence cortical excitability and cognitive functions. Caffeine associated with anodal tDCS seemed to increase MEPs and muscular strength and lower the ratings of perceived exertion when associated with anodal tDCS.”
“…with a the statistical power ranging only from 8 to 16…”
Is this suggesting that statistical power is at 8 to 16%? Statistical power is typically expressed as 1-beta, which can be expressed as a fraction (0 to 1) or as a %.
We apologize for this typo. The "%" was erroneously canceled during our internal pre-submission revisions.
5.1.Data analysis and results presentation
A point which is made implicitly with the display of Figure 1A, but should be made explicit, is the value of presenting data from all participants as a scatterplot. This allows a quick visual representation of inter-individual variability (see https://doi.org/10.1371/journal.pbio.1002128).
“In this sense, a good practice would be to always report the percentage of responders/not-responders.”
I’m in two minds on this point. On the one hand, I agree that this additional information may help contextualise some of the variability in tDCS effects. However, simple reporting of responders/non-responders fails to recognise the variability inherent in measuring a value from a participant for an outcome of interest. Using the criterion that a >50% change equates to being a ‘responder’, is a participant whose performance increases from 100 to 151 (a change of 51%) a true example of a ‘responder’ or at best a partial responder with the benefit of a positive contribution from measurement error? This issue remains true regardless of the criterion used to define ‘response’. Resolving the issue using a cluster analysis is only feasible for large samples, otherwise the solutions to these analyses are close to arbitrary median splits.
We thank the reviewer for raising these critical points. We combined these two suggestions to highlight the importance of using graphic presentation to provide detailed information concerning real data distribution. Accordingly, we added the following passage: “The way responders and non-responders have been defined is a central problem. Indeed, the division is typically based on arbitrary criteria, such as a >50% pre-post change, median split, etc. A recommendable statistical approach to identify naturally occurring subgroups that may be present in the whole sample is cluster analysis. Cluster analysis is an analytical data-driven subgrouping method that separates data based on patterns or trends ([195]; see [28] for a recent revision on cluster analysis applied to NIBS). However, this approach is feasible only for large samples. A graphical presentation of results can be of help in summarizing data distribution. Indeed, many studies (and ours among them) use bar and line graphs to show findings, typically presenting mean/median values and standard errors/deviations. However, the use of scatterplots (as in Figure 1A), dot plots, etc., allow readers (and authors) to have a clearer picture of data distribution [196], [197]. The different methods (responders vs. non-responders analyses, cluster analyses, data presentation) entail a reconsideration of null results at a group level.
“Investigating the stimulation intensity-dependent effects of tDCS on motor cortex excitability in healthy subjects, Batsikadze and colleagues [222] found a non-linear effect.”
This has also been hinted at in healthy subjects with prefrontal stimulation, showing effect sizes that are greater at 1mA than 2mA (https://doi.org/10.1016/j.brs.2018.01.003)
We thank the reviewer for suggesting this work that we added to the manuscript’s new version: “Non-linearity has also been found outside the motor regions. For example, Nikolin and colleagues applied bihemispheric tDCS over the dorsolateral regions (anode over the left, cathode over the right) during a visual 3-back working memory task. Participants received 15 minutes of bihemispheric frontal stimulation at different current intensities (1 mA, 2 mA, and three different sham conditions: 0.034, 0.016, 0 mA). Although behavioral effects did not differ in the stimulation condition, EEG activity suggested a difference between the 0 mA sham condition compared to the 1, 2, and 0.034 mA conditions, with the more significant effect sizes reported for the 1 mA condition.”
“Regarding tDCS timing, as the 4 and 5 pointed out, according…”
It’s unclear to me who the ‘4’ and ‘5’ are? Is this a citation?
We amended our typo, 4 and 5 refer to the number of the tables.
Minor text issues
“However, opposite conclusions were offered by Unal and colleagues [66], how failed to find a correlation…”
Should read ‘who’
“It is possible that nicotine assumption might affect neuroplasticity and improve cognitive performance…”
I believe this should state ‘nicotine consumption’.
“…typically assumed by patients with cardiac disease or high blood pressure…”
As above, this should state ‘typically consumed’.
“Similarly, Serotonin Selective serotonin reuptake inhibitors (SSRIs) altered the after-effects of tDCS…”
First instance of ‘serotonin’ should be deleted.
“4.Contextual and Experimental features as a source of inter-individual variability”
The formatting for this section is in italics, unlike the remainder of the manuscript body.
“Converging evidence indeded suggested…”
Typo
“Of course, a preliminar suggestion to be made regards the appropriate sample size and selection.”
Typo – should state ‘preliminary’
We thank the author for her/his suggestion. We amended the suggested typos.
Reviewer 3 Report
This is a very interesting comprehensive review investigating individual factors influencing tDCS response. The reviewer has a couple of comments/points to address before acceptance:
- In Table S1 authors report percentage of responders per study. In many of these study the number of none-responders seems to be above chance level. It might therefore be a bit dangerous to conclude that the low response rate might only be based on inter-individual differences. Many of the studies summarized in this table are not sham-controlled, so there might have been responses due to different susceptibility to placebo response, e.g.. The authors might discuss factors that might have led to higher response rates here rather than inter-individual variability, e.g. experimental design, statistical approach, (chance?).
- It is nice the authors discuss differences in induced E-field based on subject factors, however it should be stressed here that the magnitude of the induced field is an estimation. In this respect it might be helpful to discuss invasive studies measuring tDCS effects.
- One good way to validate the influence of subject factors and investigate "true" tDCS effects might be to perform a reliability study aiming for a sample that is rather homogeneous in terms of stable factors, keeping variable factors constant and adjusting experimental features. These factors might then systematically manipulated to estimate the influence of these factors. The authors might consider this point as recommendation for the discussion.
- Minor typos should be corrected.
Author Response
We thank the Reviewer for her/his revisions. We closely considered the Reviewer's suggestions and revised the draft accordingly.
Revisions are yellow highlighted in the manuscript to make the changes easier to identify.
Detailed answers to the Reviewer are listed below.
This is a very interesting comprehensive review investigating individual factors influencing tDCS response. The Reviewer has a couple of comments/points to address before acceptance.
We thank the Reviewer for her/his appreciation.
- In Table S1 authors report percentage of responders per study. In many of these studies the number of none-responders seems to be above chance level. It might therefore be a bit dangerous to conclude that the low response rate might only be based on inter-individual differences. Many of the studies summarized in this table are not sham-controlled, so there might have been responses due to different susceptibility to placebo response, e.g., The authors might discuss factors that might have led to higher response rates here rather than inter-individual variability, e.g., experimental design, statistical approach, (chance?).
We thank the Reviewer for raising this important issue, which was deepened in the introduction as follows:
"Table S1 in the Supplementary materials summarizes these works, including responders vs. non-responders' categorization, analyses, and results description. The included articles reported a significant percentage of non-responders, varying across the studies. We want to stress that the low response rate might not be based only on inter-individual differences. Conversely, discrepancies can arise from experimental features: stimulation protocols (tDCS duration and intensity, TMS intensity, the interval between tDCS stimulation and MEPs recording, the presence/absence of a sham condition), statistical procedures (e.g., cluster analysis vs. pre-post change, median split, etc.), and the number of participants included."
- It is nice the authors discuss differences in induced E-field based on subject factors, however it should be stressed here that the magnitude of the induced field is an estimation. In this respect it might be helpful to discuss invasive studies measuring tDCS effects.
We thank the Reviewer for giving us the opportunity of discussing this issue. We now added in the supplementary materials an additional table (Table S7) summarizing five studies that we found that investigated induced electric fields in in-vivo human studies. Moreover, we discussed this point in the main text as follows:
"However, two crucial points should be discussed considering electric field modeling. First, they are only an estimate of the current distribution in the brain. Indeed, many studies with in vivo or in vitro direct measurement of DC effects were performed on animals, but typically intensities are not comparable to those applied to humans. Only a few studies [222]–[226] measured in-vivo electric fields in humans, invasive methodologies are required. In-vivo human studies have been performed only in patients with implanted depth electrodes, typically due to drug-resistant epilepsy or movements disorders such as Parkinson's disease. These studies combined intracerebral recordings during stimulation delivered through tDCS [224] or tACS [222], [223], [225], [226] (see Table S7 for studies features and results description). The studies have the great value of demonstrating that transcranial electric stimulation delivered over the scalp can significantly reach also deep brain structures. Moreover, some of these studies compared estimated electric fields provided by modeling with the measured electric fields [223], [226], providing empirical support to the models' accuracy. Secondly, future research should address the effect of electric fields' strength on the behavioral outcome to clarify how morphological differences can translate into different tDCS-induced neurophysiological and behavioral effects."
- One good way to validate the influence of subject factors and investigate "true" tDCS effects might be to perform a reliability study aiming for a sample that is rather homogeneous in terms of stable factors, keeping variable factors constant and adjusting experimental features. These factors might then systematically manipulate to estimate the influence of these factors. The authors might consider this point as recommendation for the discussion.
We added this point in the discussion as follows:
Crucially, to investigate the influence of inter-individual differences in tDCS effects, authors should perform studies in which the sample is homogeneous in stable factors. In contrast, the experimenters should systematically manipulate one variable factor at a time to deepen the knowledge on the influence of each element.
- Minor typos should be corrected.
We amended typos throughout the manuscript and the supplementary materials.
Round 2
Reviewer 1 Report
I full understand that authors did not aim for systematic review but the narrative on (which is now clear in the title), and I ma fully aware how PRISMA applies to this type of review.
However my main concern was not technical, but rather essential to any type of review - how do authors assure that the review is not biased. Without explicit search strategy it could be argued that some works are intentionally omitted. Therefore, I can not reccomend publishing the work before this critical issue is addressed.
a quck PubMed search (https://pubmed.ncbi.nlm.nih.gov/?term=individual+diffrences+tdcs) shows that the some works are not covered:
Katz B, Au J, Buschkuehl M, Abagis T, Zabel C, Jaeggi SM, Jonides J. Individual Differences and Long-term Consequences of tDCS-augmented Cognitive Training. J Cogn Neurosci. 2017 Sep;29(9):1498-1508. doi: 10.1162/jocn_a_01115
Falcone B, Wada A, Parasuraman R, Callan DE. Individual differences in learning correlate with modulation of brain activity induced by transcranial direct current stimulation. PLoS One. 2018 May 21;13(5):e0197192. doi: 10.1371/journal.pone.0197192
Therefore I would strongly suggest having an explicit search strategy and of course adding all other relevant works on top of that.
Author Response
We thank the Reviewer for her/his suggestions that we carefully considered. Although we fully understand the Reviewer's concern, the only solution to this would be to follow a systematic strategy in the paper search, likewise following PRIMA guidelines, which would imply changing the structure of our work entirely, switching from a narrative to a systematic review. However, this was not our purpose, and, anyway, it cannot be done in the four days allowed for the revision since it will require months of work, given the breadth of the topics covered in the draft.
The Reviewer suggested that she/he ran a "quick Pubmed search" and found some articles that are not in our review. This is fine because narrative reviews do not pretend to cover all the available literature. We found the work by Katz and colleagues (2017), but we decided not to include it, given that all the included articles combining tDCS with tasks were single sessions or crossover studies. Indeed, our search strategy was similar to the one suggested by the Reviewer, combining “tDCS” AND “interindividual differences” or “tDCS” AND “interindividual variability”. However, we do not find it appropriate to report these strategies in the manuscript. It would generate the expectancy in the reader that we covered all the pertinent studies as we conducted a systematic review, which is not the case. We agree that this can be a limitation and acknowledge it through the draft. Actually, this is a limitation inherent to any narrative review. On the other side, there is a difference between not covering the literature of a topic exhaustively and intentionally omitting relevant literature (although we found difficult to find a rationale for that). We believe that the wide range of authors cited and reported in the tables can guarantee the absence of a bias in citing only certain authors. Conversely, as previously asked, if the Reviewer is aware of crucial articles that we have unintentionally omitted, we would be thankful to add them, believing it will ameliorate our narrative review. Again, without following a systematic strategy, the risk of omitting relevant work is inherent to any narrative review.
Reviewer 3 Report
The authors have addressed all of my comments satisfyingly. I recommend publication.